# Neutrophils dominate in opsonic phagocytosis of *P. falciparum* blood-stage merozoites and protect against febrile malaria

Asier Garcia-Senosiain [1,2,10], Ikhlaq Hussain Kana [1,2,10], Subhash Singh [3,9 ✉], Manoj Kumar Das[4], Morten Hanefeld Dziegiel [5,6], Sanne Hertegonne [2,7], Bright Adu [8 ✉] & Michael Theisen [1,2 ✉]

Antibody-mediated opsonic phagocytosis (OP) of *Plasmodium falciparum* blood-stage merozoites has been associated with protection against malaria. However, the precise contribution of different peripheral blood phagocytes in the OP mechanism remains unknown. Here, we developed an in vitro OP assay using peripheral blood leukocytes that allowed us to quantify the contribution of each phagocytic cell type in the OP of merozoites. We found that CD14$^{++}$CD16$^-$ monocytes were the dominant phagocytic cells at very low antibody levels and Fc gamma receptor (FcγR) IIA plays a key role. At higher antibody levels however, neutrophils were the main phagocytes in the OP of merozoites with FcγRIIIB acting synergistically with FcγRIIA in the process. We found that OP activity by neutrophils was strongly associated with protection against febrile malaria in longitudinal cohort studies performed in Ghana and India. Our results demonstrate that peripheral blood neutrophils are the main phagocytes of *P. falciparum* blood-stage merozoites.

[1] Department for Congenital Disorders, Statens Serum Institut, Copenhagen, Denmark. [2] Centre for Medical Parasitology at Department of International Health, Immunology and Microbiology, University of Copenhagen and Department of Infectious Diseases, Copenhagen University Hospital, Rigshospitalet, Copenhagen, Denmark. [3] Indian Institute of Integrative Medicine, Jammu, India. [4] National Institute of Malaria Research, Field Unit, Ranchi, Jharkhand, India. [5] Blood Bank KI 2034, Department of Clinical Immunology, Copenhagen University Hospital, Copenhagen, Denmark. [6] Department of Clinical Medicine, University of Copenhagen, Copenhagen, Denmark. [7] Ghent University, Ghent, Belgium. [8] Noguchi Memorial Institute for Medical Research, College of Health Sciences, University of Ghana, Accra, Ghana. [9] Present address: Consytel Life Sciences Pvt. Ltd., Hyderabad, India. [10] These authors contributed equally: Asier Garcia-Senosiain, Ikhlaq Hussain Kana. ✉email: subhash0974@gmail.com; BAdu@noguchi.ug.edu.gh; mth@ssi.dk

Blood-stage *Plasmodium spp.* infection is responsible for the clinical manifestations of malaria and is characterized by intraerythrocytic replication of asexual parasites and the release of free merozoites into circulation to invade new erythrocytes[1]. Free merozoites are more vulnerable to immune factors, including antibodies, complement, and leukocytes that protect the host (reviewed in[2–4]). Passive transfer of IgG antibodies from individuals with naturally acquired immunity (NAI) against malaria to *Plasmodium falciparum*-infected patients could control parasiteamia and alleviate fever[5,6]. Malaria-specific IgG antibodies may exert their biological function either by directly preventing merozoite invasion of erythrocytes[7,8] or in co-operation with immune effector cells through cell-surface Fc gamma receptors (FcγR)[9,10]. Engagement of FcγRs triggers potent immune cell mechanisms resulting in the release of soluble factors[11] and/or phagocytosis[12–14] leading to parasite killing. Of these mechanisms, the opsonic phagocytosis (OP)[12–14] has by far been the most consistent correlate of protection against malaria in sero-epidemiological studies. However, all the bioassays used in studying these mechanisms have mostly relied on the use of preselected cell types, such as monocytes[11,15] or neutrophils[11,16], or cell lines belonging to definite lineages[13,14,16,17], which may not clearly depict the peripheral blood environment. In a recent study, peripheral blood neutrophils in a whole-leukocyte assay were shown to play a prominent role in the clearance of *P. falciparum* sporozoites demonstrating their role in pre-erythrocytic stage malaria immunity[18]. The role of the different peripheral blood phagocytes in opsonic phagocytosis of blood-stage *P. falciparum* merozoites and their overall contribution to blood-stage malaria immunity is yet to be described using such unbiased in vitro bioassays. Here, using an in vitro phagocytosis assay, which included all peripheral blood leukocytes (PBL) (in proportion to their normal composition in peripheral blood), we investigated the role of different cell types and their respective FcγRs in IgG-mediated phagocytosis of blood-stage *P. falciparum* merozoites. Using our assay, we identified neutrophils and monocytes as the two major phagocytic cell types, and that the levels of antibodies deposited on the merozoite surface (as indicated by IFA titers) determined which cell type played the dominant phagocytic role. Higher levels of antibody deposition on merozoites (as indicated by higher IFA titers), stimulated neutrophils to play a dominant phagocytic role, while with lower levels of antibody deposition on merozoites (as indicated by lower IFA titers) the monocytes played the dominant phagocytic role. Further, using the phagocytosis assay we tested the association between IgG-mediated OP of blood-stage *P. falciparum* merozoites and protection against febrile malaria in two different longitudinal cohort studies performed in Ghana and India.

## Results

### Neutrophils are the major cell types involved in IgG-mediated phagocytosis of *P. falciparum* blood-stage merozoites.

To investigate IgG-mediated phagocytosis of *P. falciparum* blood-stage merozoites, PBLs from whole blood samples were used to ensure that the different leukocytes (lymphocytes, monocytes, and granulocytes) were present in their natural physiological proportions in the phagocytosis assays.

The Danish donor PBLs used contained approximately 6% monocytes, 59% neutrophils, 32% lymphocytes, and 3% eosinophils and basophils (Fig. 1a). For IgG source, we used de-complemented human plasma samples at 1:8000 dilutions from malaria immune individuals (immune plasma [IP]); or malaria nonexposed Danish donors (nonimmune plasma [NIP]) as in our initial experiments complement did not play a significant role in the OP of merozoites. PBLs that phagocytosed ethidium bromide-

stained merozoites were quantified by flow cytometry. We found that although both the neutrophils and monocytes were active in the phagocytosis of merozoites, neutrophils were more active despite the antibody type (i.e., IP or NIP) used to opsonize merozoites (Fig. 1b). Importantly, median merozoite-phagocytosis by neutrophils in presence of IP was markedly enhanced compared to NIP (68-fold) and no plasma (unopsonized [UOP]) (95-fold) controls. Moreover, opsonization with IP led to a substantial shift (Wilcoxon signed-rank test, $P = 0.002$) in the relative distribution in the phagocytic fraction with neutrophils constituting 94% and monocytes only 6% as compared to 70 and 30%, respectively, in the unopsonized condition (Fig. 1b, lower panel). This indicates that the opsonic phagocytosis of blood-stage merozoites by the neutrophils may be a potent immune protective mechanism against blood-stage malaria infection. Next, to estimate the capacity of each phagocyte to take up UOP or antibody-opsonized merozoites, phagocytosis indices (%) were calculated for both the neutrophils and monocytes (Fig. 1c). The median percentage of merozoite-phagocytosis by neutrophils was found to be significantly higher (1.4-fold) than that of monocytes (Wilcoxon signed-rank test, $P = 0.002$). Merozoite-phagocytosis by both the neutrophils and monocytes were abolished by pretreatment of cells with cytochalasin D (Cyt D), a reliable inhibitor of phagocytosis (Fig. 1c). To determine the kinetics of merozoite OP by PBLs, we investigated the time course for the uptake of opsonized merozoites by each phagocyte (Fig. 1d). IP-opsonized merozoites were avidly internalized by both the neutrophils and monocytes with a mean of 85% of neutrophils and 84% of monocytes phagocytosing merozoites by the first 1 min (Fig. 1d). Merozoite-phagocytosis could be abolished by Cyt D, demonstrating that opsonized merozoites become actively ingested shortly after contact with the respective phagocytes (Fig. 1d).

Of the monocyte subsets, we found that classical (CD14$^{++}$ CD16$^-$) monocytes contributed most to the merozoite-phagocytosis in our assay compared to nonclassical (CD14$^+$ CD16$^+$) and intermediate (CD14$^{++}$CD16$^+$) monocytes (Supplementary Figure 1a and b). Interestingly, the less abundant CD14$^+$CD16$^+$ and CD14$^{++}$CD16$^+$ subsets were the most efficient phagocytes on a per cell basis (Supplementary Figure 1c).

Taken together, we conclude here that neutrophils are the main phagocytes of antibody-opsonized merozoites in peripheral blood while classical monocytes dominate the monocytic contribution due to their relatively higher numbers although not being the most efficient phagocytes on per cell basis.

### The degree of antibody opsonization influences merozoite-phagocytosis by neutrophils and monocytes.

To further characterize merozoite-phagocytosis by PBLs, we investigated whether the degree of antibody opsonization influenced phagocytosis. The merozoites were opsonized with increasing dilutions of immune plasma ($n = 21$) and added separately to PBLs from a single donor. There was a clear dose-dependent effect on merozoite-phagocytosis by neutrophils and monocytes demonstrating that phagocytosis depends on the amount of antibody deposited on the merozoite surface for which we relied on quantification through IFA titers (Fig. 2a). Overall, neutrophils were the main phagocytes of antibody-opsonized merozoites, thus confirming the above results. Interestingly, there seems to be a shift in the relative contribution of neutrophils and monocytes to the overall phagocytosis with decreasing amounts of opsonizing antibodies observed at increasing plasma dilution levels (Fig. 2a). At high antibody levels neutrophils are the main phagocytes, while the contribution of monocytes in merozoite-phagocytosis became more prominent at lower levels of opsonizing antibodies at higher

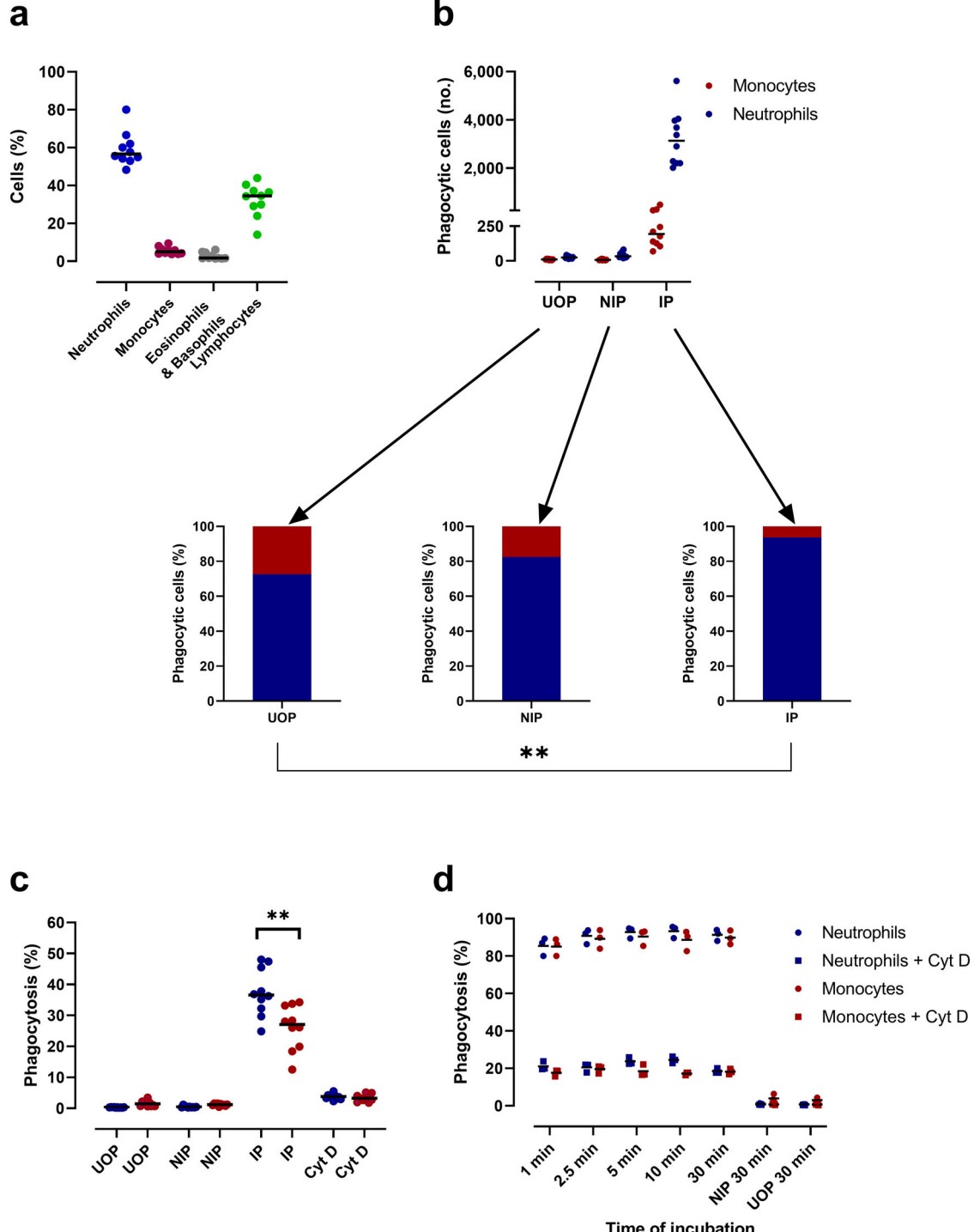

**Fig. 1 Malarial antibodies induce merozoite-phagocytosis by neutrophils and monocytes. a** Percentage of individual leukocytes in PBLs from Danish blood donors ($n = 10$) was enumerated by flow cytometry. Horizontal lines represent median values. **b** From each donor, $6 \times 10^4$ PBLs were incubated with unopsonized merozoites (UOP), merozoites opsonized with a pool of nonimmune plasma (NIP), or with immune plasma (IP) for 30 min at 37 °C for phagocytosis to occur. Cells were stained with anti-CD14, CD16, CD45, and CD66b antibodies and analyzed by flow cytometry to enumerate ethidium bromide (EtBr)-positive cells. The mean number (no.) of neutrophils (blue) and monocytes (red), which ingested merozoites are shown in the upper panel, and the percent contribution of neutrophils and monocytes to overall PBL phagocytosis is shown in lower panels for the given opsonized/unopsonized condition. **c** Mean percentage of monocytes or neutrophils in PBL preparations that phagocytosed merozoites in the presence of NIP, IP, or UOP condition. The percent phagocytosis of IP-opsonized merozoites is also shown for cytochalasin D (Cyt D) treated PLBs. **d** Kinetics of IP-opsonized merozoite uptake by monocytes and neutrophils present in Cyt D pretreated or untreated PBLs. Phagocytosis was stopped at indicated time points, and percent phagocytosis by the neutrophils and monocytes is expressed as the mean of triplicates. Data are represented from one of the two independent assays. Results were obtained with PBLs from 10 different Danish blood donors (**a–c**). P values were determined by Wilcoxon signed-rank test **b** and **c**. Asterisks represent P values (**$P < 0.01$).

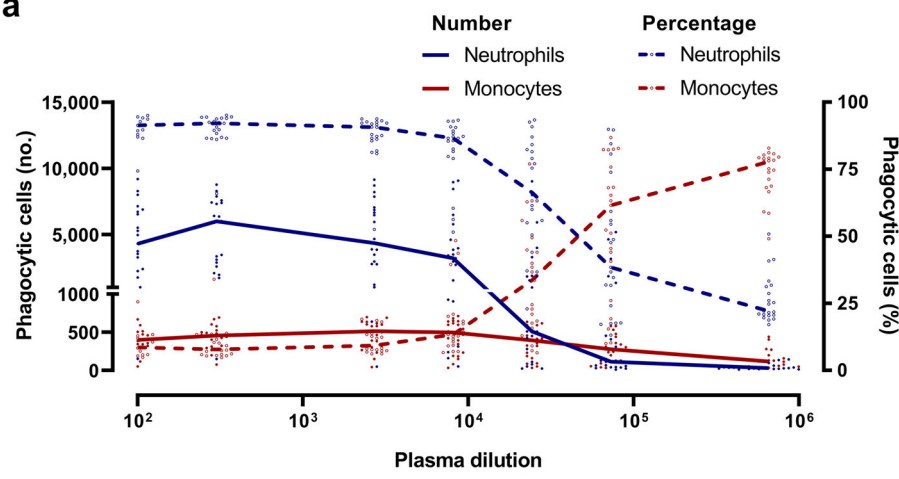

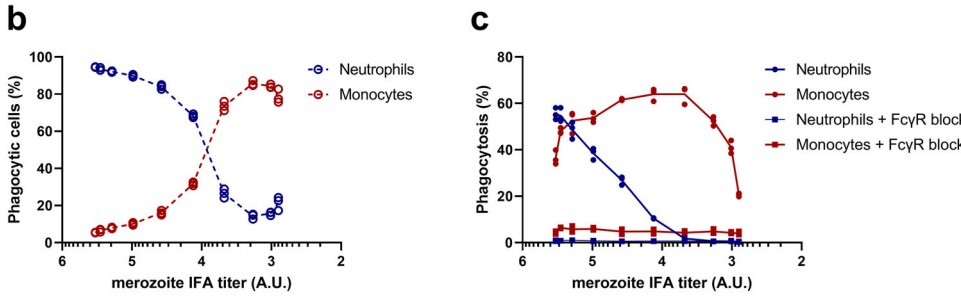

**Fig. 2 Merozoite-phagocytosis by PBLs depends on the degree of antibody opsonization.** Merozoites were opsonized with serial dilutions of individual plasma ($n = 21$) from Ghanaian children **a** or a pool of the same plasma **b** and **c** and incubated with PBLs ($6 \times 10^4$ per well) for 30 min at 37 °C for phagocytosis to occur. Cells were stained with anti-CD14, CD16, CD45, and CD66b antibodies and analyzed by flow cytometry to enumerate EtBr-positive cells. **a** Left y-axis shows the median number (no.) of EtBr-positive neutrophils (blue solid line) and monocytes (red solid line) present in PBLs in the presence of individual plasma samples ($n = 21$). Right y-axis shows the median percentage of EtBr neutrophils (dotted blue line) and monocytes (dotted red line) in EtBr-positive PBL fraction in the presence of individual plasma samples ($n = 21$). **b** Mean percent of EtBr-positive neutrophils and monocytes in EtBr-positive PBL fraction in the presence of pooled plasma is plotted against the amount of antibody deposited on the merozoite surface as quantified by the merozoite-immunofluorescence assay (IFA). **c** Mean percentage of EtBr-positive neutrophils and monocytes present in PBL preparations either untreated or pretreated with FcγR blockers in the presence of pooled plasma is plotted against merozoite-IFA. The experiments in **b** and **c** were performed in triplicates.

plasma dilutions (generalized estimating equation, $P < 0.0001$). Next, the dose-titration experiment was repeated with the pool of the immune plasma tested above. The phagocytic activity of PBLs was plotted against the amount of malarial IgG deposited on the merozoite surface as determined by merozoite-IFA (Fig. 2b). As expected, the neutrophils were the main phagocyte at high antibody concentrations (merozoite-IFA titer $>10^4$), while monocytes became the dominant phagocyte at lower antibody concentrations corresponding to <10% saturation of the merozoite surface (Fig. 2b). Nonimmune plasma promoted low (<10%) phagocytosis activities at all antibody concentrations. We observed that with higher dilutions of plasma samples corresponding to change in IFA titers from $10^5$ to $10^3$ (A.U.), there was a distinct change in the phagocytic activities of neutrophils and monocytes (Fig. 2b). While at higher IFA titers $>10^5$ neutrophils were observed to be the dominant phagocytic cells, with higher plasma dilutions resulting in IFA titers $<10^4$ the phagocytic activity of neutrophils decreased linearly to near zero on a per cell basis (Fig. 2c). Interestingly, the monocytes which displayed a less dominant phagocytic role at higher IFA titers $>10^5$, displayed a dominant phagocytic role at higher plasma dilutions resulting in IFA titers $<10^4$, and further retained robust phagocytic activity on a per cell basis even at a higher dilution of $10^3$ (Fig. 2c). The shift in the relative contribution of phagocytes is explained by monocytes

being the most efficient phagocytic cell at this low antibody concentration. Importantly, merozoite-phagocytosis by both of these cell types could be abolished by the blocking of all surface-exposed FcγRs using a combination of anti-CD16, anti-CD32, and anti-CD64 antibodies prior to their use in the phagocytosis assay, demonstrating that phagocytosis is FcγR-dependent at all IgG concentrations tested (Fig. 2c).

Collectively, these findings suggest that opsonization of merozoites leads to prominent phagocytosis by monocytes at low IgG densities, whereas neutrophils are the dominant phagocytes at higher IgG densities, as reflected by the merozoite-IFA titer.

**IgG-mediated merozoite-phagocytosis by neutrophils and monocytes depend on engagement of distinct FcγRs.** To dissect the role of FcγR signaling in merozoite-phagocytosis by blood phagocytes, we investigated PBLs from 19 blood donors. We used anti-CD16 (FcγRIIIB), anti-CD32 (FcγRIIA), and anti-CD64 (FcγRI) antibodies to block individual FcγRs. Median merozoite-phagocytosis activity by neutrophils was reduced by 90% as compared to isotype control when the interaction of the merozoite-antibodies with FcγRIIIB was blocked (Friedman test, $P = 0.0009$) (Fig. 3a). Phagocytosis by neutrophils was unaffected

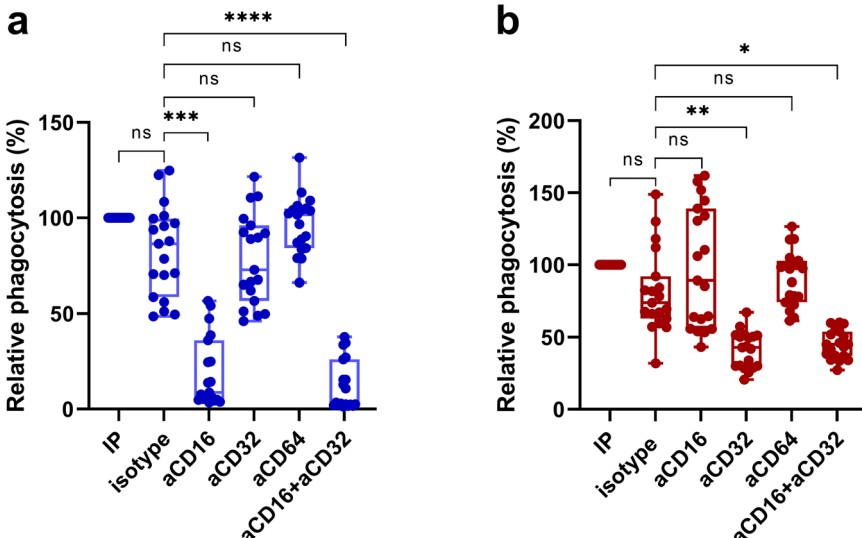

**Fig. 3 Phagocytosis of antibody-opsonized merozoites depends on FcγRIII in neutrophils and FcγRII in monocytes.** The PBLs ($n = 19$) were treated with 10 µg/ml of antibodies against CD16, CD32, CD64, or an isotype control for 30 min and then incubated with IP-opsonized merozoites for phagocytosis to occur. Graphs show the relative phagocytosis (%) of specific FcγR blocker treated neutrophils **a** and monocytes **b** using the untreated homologous phagocytes as reference. Boxes indicate the median and interquartile range. Whiskers mark the minimum and maximum values. *P* values were determined by the Friedman test and Dunn's multiple comparisons test. Asterisks represent *P* values (*$P < 0.05$; **$P < 0.01$; ***$P < 0.001$; ****$P < 0.0001$).

by the addition of anti-CD32 or anti-CD64 antibodies suggesting that FcγRIIIB is the main receptor for phagocytosis by neutrophils (Fig. 3a). Interestingly, simultaneous blocking of both the FcγRIIIB and FcγRIIA in neutrophil preparations further reduced the median merozoite-phagocytosis activity by 61% (Wilcoxon signed-rank test, $P = 0.0010$) suggesting that these receptors complement each other in IgG-mediated phagocytosis by neutrophils under the conditions tested here.

The data also demonstrated that blocking antibodies specific for CD32 inhibited median merozoite-phagocytosis by monocytes by about 42% compared to isotype control (Friedman test, $P = 0.0015$). Blocking antibodies specific for CD16 and CD64 had no such effects (Fig. 3b) suggesting that FcγRIIA but not FcγRIIIB and FcγRI is necessary for merozoite-phagocytosis by monocytes.

Altogether, these data suggest that FcγRIIIB (in synergism with FcγRIIA) is a major mediator of merozoite-phagocytosis by neutrophils while FcγRIIA alone seems to perform this function in monocytes.

**Acquisition of IgG capable of mediating opsonic phagocytosis by PBLs is associated with protection against febrile malaria.** To investigate the association between antibody-mediated merozoite-phagocytosis by human PBLs and protection against febrile malaria, we used clinical data and plasma samples ($n = 140$) from a well-established longitudinal cohort survey (LCS) performed in Ghanaian children[19] (Supplementary Table 1). The PBLs used here in phagocytosis assays were obtained from a Danish blood donor. A wide range of phagocytic responses was observed for both neutrophils (median number of phagocytosing cells 1154; interquartile range (IQR), 65–2717), and monocytes (median number of phagocytosing cells, 578; IQR, 147.3–1004), and phagocytosis levels with neutrophils was significantly higher compared to monocytes (Wilcoxon matched-pairs signed-rank test, $P < 0.0001$) (Fig. 4a). For most of the plasma samples, neutrophils were the main phagocyte of merozoites due to their greater numbers and apparently stronger phagocytic efficiency (Fig. 4a). Overall, phagocytic activities for neutrophils (median phagocytosis, 24.6%; IQR, 1.6–56.7%) and monocytes (median

phagocytosis, 22.1%; IQR, 5.4–38%), respectively increased with the age of children, reflective of cumulative exposure to malaria. This was statistically significant for both cell types; however, the stronger association was observed for neutrophils (Odds ratio [OR] = 20.4; 95% CI = 4.3–96.7; $P = 0.0001$) compared to monocytes (OR = 6.8; 95% CI = 2.5–18.1; $P = 0.0001$).

Next, children were categorized into two equal groups based on the median phagocytosis value for the respective phagocytes to examine the risk of symptomatic malaria for each phagocytosis-responder group using the Cox-regression models to calculate hazard ratios (HRs). Children in the high phagocytosis group for both neutrophils (unadjusted [uHR] = 0.48; 95% CI = 0.31–0.75; $P = 0.001$) and monocytes (uHR = 0.51; 95% CI = 0.33–0.80; $P = 0.003$) had a significantly lower risk of febrile malaria over a 42-week follow-up period compared to those with low-level responses (Fig. 4b). The strength of protective association was similar between neutrophils and monocytes. Adjusting the Cox-regression models for age, as it is correlated with parasite exposure and febrile malaria outcome in this cohort[20], marginally influenced the magnitude of protective associations (Fig. 4b). Next, we repeated the phagocytosis assays using highly purified neutrophils (>99%; obtained from a Danish blood donor) and merozoites opsonized with the same Ghanaian plasma samples ($n = 140$)[19] (Supplementary Table 1). Children in the high purified neutrophil-phagocytosis group had a significantly (uHR = 0.43; 95% CI, 0.28–0.68; $P < 0.001$ and age-adjusted (aHR) = 0.47; 95% CI, 0.30–0.75; $P = 0.001$) higher probability of remaining free of malaria during the study period than those in the low purified neutrophil-phagocytosis group. Thus, supporting the importance of neutrophils in parasite-killing mechanisms.

To further extend these findings and to strengthen the finding that neutrophils are important for NAI against malaria, we comparatively assessed plasma samples ($n = 121$) in PBL-phagocytosis assay and used clinical data from a LCS conducted in India[21,22] (Supplementary Table 1). In general, neutrophils in PBL preparations (obtained from a Danish blood donor) showed higher phagocytosis activity than monocytes (Fig. 4c). Cox proportional-hazard models used to determine estimates of protection confirmed that subjects in high phagocytosis group

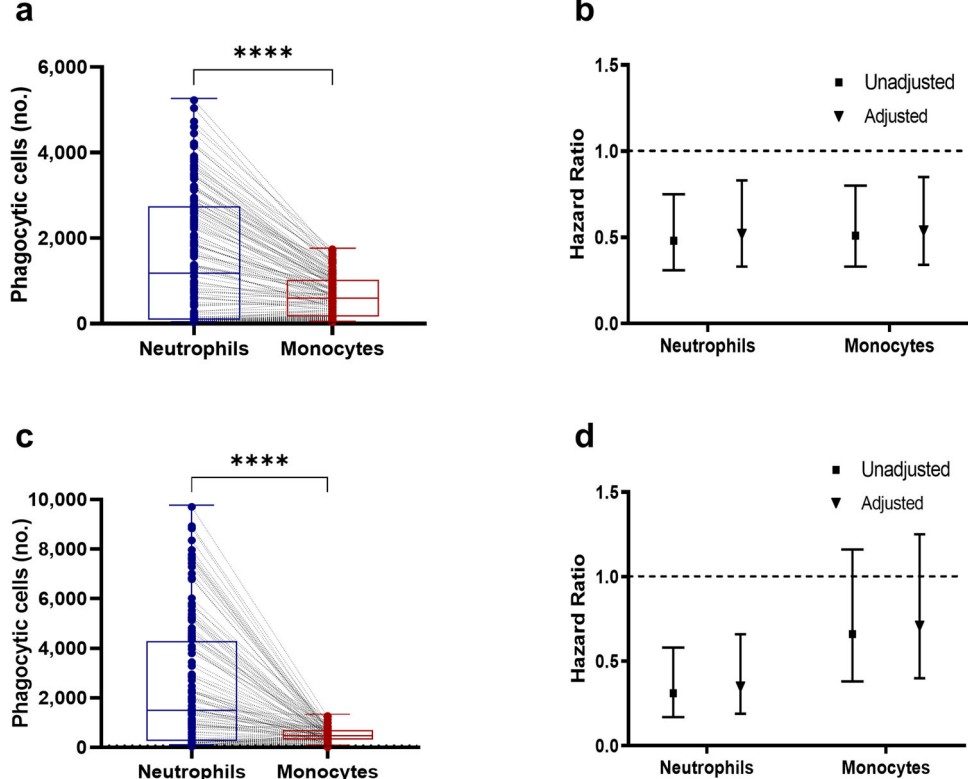

**Fig. 4 Neutrophil-phagocytosis of merozoites is dominant compared to monocytes and predicts protection against febrile malaria.** Paired aligned dot plots coupled to box and whiskers plots showing the number of EtBr-positive neutrophils and monocytes present in PBLs from Danish blood donors which were used as effector cells to test the phagocytosis mediating ability of antibodies present in Ghanaian **a** and Indian **c** cohort plasma samples. Both the Ghanaian children ($n = 140$; protected = 59, susceptible = 81) **b** and the Indian participants ($n = 121$; protected = 71, susceptible = 50) **d** were categorized into two equal groups based on the median phagocytosis values obtained with neutrophils or monocytes, and the risk of encountering febrile malaria during the follow-up periods was calculated with Cox-regression models comparing those with high versus low (reference group) phagocytosis. Values represent age-adjusted (filled triangles) and unadjusted (filled squares) hazard ratios with 95% confidence intervals. Boxes indicate the median and interquartile range. Whiskers mark the minimum and maximum values. *P* values were determined by Wilcoxon signed-rank test **a** and **c**. Asterisks represent *P* values (****$P < 0.0001$).

for neutrophils (aHR = 0.35; 95% CI = 0.19–0.66; $P = 0.001$) but not monocytes (aHR = 0.71; 95% CI = 0.40–1.25; $P = 0.23$) had a significantly lower risk of symptomatic malaria over the follow-up period compared to those with low-level responses (Fig. 4d).

Collectively, these findings reinforce the importance of merozoite-phagocytosis in NAI in both Africa and Asia and points to a particularly important and dominant role of peripheral blood neutrophils.

**Relationship between antibody-mediated merozoite-phagocytosis by PBLs and antibody-dependent cellular inhibition activity in Ghanaian children.** The antibody-dependent cellular inhibition (ADCI) assay measures the parasite-killing capacity of antibodies in collaboration with human monocytes. We, therefore, compared the phagocytosis activity of PBLs with the ADCI activity previously reported for the same IgG preparations ($n = 85$) used here[23]. There was no correlation between merozoite-phagocytosis by neutrophils (Spearman $r = -0.04$; $P = 0.68$) or monocytes (Spearman $r = -0.06$; $P = 0.5$) and the ADCI activity suggesting that phagocytosis by both the monocytes and neutrophils are distinct mechanisms and different from that of ADCI (Fig. 5).

## Discussion
Here we have taken an unbiased approach to investigate the role of different blood leukocytes in antibody-mediated phagocytosis

of *P. falciparum* extracellular merozoites. We have focused on asexual blood-stage parasites because they are clinically important and responsible for all symptoms associated with *P. falciparum* malaria. Using an in vitro phagocytosis assay, which included physiological proportions of all peripheral blood leukocytes from healthy donors, we demonstrated that both the human neutrophils and monocytes were highly active in the phagocytosis of extracellular merozoites. We found that neutrophils were the dominant phagocyte of merozoites in our in vitro assay.

In some respects, this is not surprising, as neutrophils are the most abundant (50–70%) leukocytes in human blood-forming the first line of defence following an infection. They possess a range of antimicrobial effector functions like respiratory burst activity, which releases reactive oxygen species (ROS). ROS are highly toxic to intraerythrocytic malaria parasite development[24–26] and neutrophil-mediated antibody-dependent respiratory burst (ADRB) is associated with protection against clinical malaria[16]. However, it is quite surprising that despite such aforementioned potent antimicrobial properties, the implications of IgG-mediated phagocytosis by neutrophils in immunity against blood-stage malaria parasites have been quite understudied[27–30].

In an attempt to characterize effector functions of specific phagocytes in blood, several factors were investigated, which may potentially influence their capacity for phagocytosis of merozoites. Initially, we demonstrated that the degree of antibody opsonization was important for merozoite-phagocytosis by both

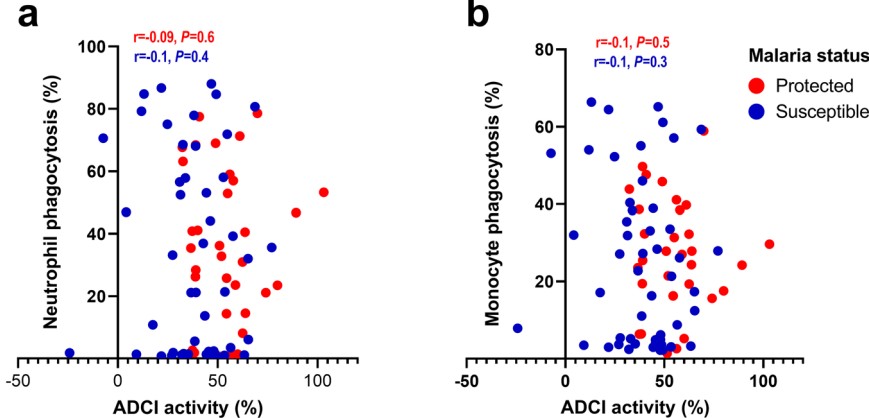

**Fig. 5 ADCI activity is not associated with merozoite-phagocytosis by neutrophils or monocytes.** Scatterplots showing the relationship between ADCI activity and merozoite-phagocytosis by neutrophils **a** and monocytes **b** in a Ghanaian cohort of children. Here, the data are presented for 85 children (protected [red dots] = 33, susceptible [blue dots] = 52) for whom both phagocytosis and ADCI activity were available. Spearman's correlation coefficient r, with associated P value is shown for both protected (red font) and susceptible (blue font) children in each plot.

the neutrophil and monocyte cell populations. However, contrary to expectations, we observed that monocytes were actively ingesting merozoites at lower antibody concentrations compared to neutrophils suggesting that the threshold of antimerozoite antibodies to mediate OP for peripheral monocytes is lower than that of peripheral neutrophils. Once the threshold of anti-merozoite antibodies for phagocytosis by neutrophils has been reached, these cells became the dominant phagocyte accounting for 94% of the phagocytic fraction. Thus, emphasizing the importance of neutrophils for the elimination of *P. falciparum* blood-stage parasites.

Since IgG antibodies mediate cellular immune effector pathways through binding to FcγRs encoded by different genes we examined the role of specific FcγRs on merozoite-phagocytosis. Human monocytes and neutrophils both express FcγRIIA with a subset of monocytes expressing FcγRIIIA while FcγRIIIB expression is exclusive to neutrophils[31]. We found that anti-CD16 antibodies efficiently blocked merozoite-phagocytosis demonstrating that FcγRIIIB is the main receptor for antibody-mediated phagocytosis by neutrophils. Interestingly, anti-CD32 antibodies enhanced the blocking activity of anti-CD16 antibodies suggesting that FcγRIIIB and FcγRIIA act in concert. FcγRIIIB is bound to the cell membrane through its glycosylphosphatidylinositol (GPI) anchor and is completely lacking a signal module or signaling capacity by itself[23]. Therefore, FcγRIIIB most likely requires help from other membrane protein(s) with signaling capacity. Some studies have suggested FcγRIIIB may act synergistically with activating receptors such the ITAM-bearing single-chain FcγRIIA for signaling[32], thus providing a plausible explanation for the synergy between anti-CD16 and anti-CD32 antibodies in merozoite-phagocytosis observed here. A recent report has further demonstrated that human neutrophils may express low levels of FcγRIIIA[33]. Whether FcγRIIIB and FcγRIIIA might cooperate in merozoite-phagocytosis by neutrophils remains to be investigated. Our blocking experiments also demonstrated that FcγRIIA was the sole receptor for mediating the phago-cytosis of merozoites by monocytes. We, therefore, hypothesize that differences in FcγR density, as well as specificity and affinity toward IgG subclasses might explain the different thresholds for merozoite-phagocytosis by neutrophils and monocytes, respectively. At low antibody concentrations (merozoite-IFA titer <$10^4$ in our PBL assay), opsonized merozoites will initially

bind to FcγRIIA, which has a 10- to 40-fold higher affinity for IgG compared to FcγRIIIB[34]. Thus, favoring FcγR activation on monocytes and phagocytosis. Higher antibody concentrations in plasma leads to a proportional change in the amount of IgG deposited on the merozoite surface thereby enhancing engagement of FcγRIIIB on neutrophils and subsequently phagocytosis. The very high density of FcγRIIIB ($\sim 1.4 \times 10^6$ receptors per cell) on neutrophils may further lead to extensive receptor clustering at the membrane[35] and enhanced activation and merozoite-phagocytosis.

To investigate the role of neutrophils in NAI against clinical malaria, we have used a well-characterized LCS to show that the neutrophil-phagocytosis activity of IgG is significantly associated with reduced risk against febrile malaria in Ghanaian children. Our finding that children whose antibodies showed merozoite-phagocytosis activities by neutrophils above the median had a 48% (effect of age accounted) reduced risk of febrile malaria compared to children with activities below the median strongly supports the notion that the neutrophils play an important role in malaria immunity. This association was observed in two inde-pendent experiments using both the PBLs and highly purified (>99%) neutrophils. These findings corroborate a recent study that showed that neutrophil opsonic phagocytosis of sporozoites is important in malaria immunity and suggest that the role of neutrophils in controlling parasitemia may transcend different stages of the parasite[18]. Since the PBL-phagocytosis assay devel-oped here measures the phagocytic activity of all leukocytes, our analysis also demonstrated that the phagocytosis promoting activity of monocytes was significantly associated with a reduced risk of febrile malaria in this Ghanaian cohort. While the role of monocytes in NAI against malaria is well established[12–14,23], we show here that antibody-mediated phagocytosis of blood-stage *P. falciparum* by neutrophils is also involved in NAI. Due to their greater numbers, we predict that neutrophils are the main effector cells that control blood-stage malaria parasite multiplication in vivo a notion, which may also hold true for pre-erythrocytic stage infections[18].

Since NAI against malaria is most likely a composite of mul-tiple antiparasite protective mechanisms we sought to compare the data obtained here with previous data from the same LCS. We found no correlation between merozoite-phagocytosis by either monocytes or neutrophils and ADCI confirming our past observation that there was no significant relationship between

phagocytosis and ADCI[20]. Collectively these data suggest that phagocytosis and ADCI are distinct parasite-killing mechanisms.

While this study has focused on FcγR-mediated phagocytosis, yet other studies have suggested that clearance of merozoites may also involve complement[36,37]. The current study suggested that complement did not enhance antibody-mediated phagocytosis of merozoites. We did not further investigate the potential role of complement-receptor(s) in merozoite clearance since this was outside the scope of the present study.

This study has identified both the monocytes and neutrophils as two major peripheral blood leukocytes, with different physiological properties, contributing to IgG-mediated phagocytosis of malaria parasites, which play role in immunity against malaria. Altogether, our results provide an important glimpse into the various facets of multiple IgG-mediated cellular responses contributing to the phenomenon of antibody-mediated NAI to malaria. The neutrophils constitute the largest myeloid cell subset, and are considered short-lived frontline responders against pathogens; however, recent evidence suggests they have the capacity to prime adaptive immune responses during phagocytosis[38]. The monocytes on the other hand appear to have comparatively lower phagocytic activity than neutrophils but are significantly longer-lived efficient antigen-presenting cells. Thus, OP of blood-stage merozoites by both cell types seem to complement each other in the development of protective immunity against malaria, and our results show that their respective IgG-mediated phagocytic activities are associated with protection against febrile malaria in field settings.

In summary, our study shows that neutrophils contribute significantly to IgG-mediated phagocytosis of blood-stage *P. falciparum* merozoites and protection against febrile malaria. These findings further elucidate neutrophil antibody-mediated immune mechanisms against blood-stage malaria parasites.

## Methods

**Ethics statement**. Ethical approval for Danish blood donor samples was given by the Scientific Ethics Committee of Copenhagen and Frederiksberg, Denmark. Samples from anonymous Danish blood donors (18–60 years of age) obtained for control purposes at Copenhagen University Hospital were used. These individuals are resident of central Copenhagen and provided written consent to have a small portion of their blood stored anonymously and used for research purposes. All data were analyzed anonymously.

The Ghanaian LCS was approved by the Institutional Review Board of Noguchi Memorial Institute for Medical Research of the University of Ghana, Accra, Ghana, and the Indian LCS was approved by the Institutional Ethics Committee of the National Institute of Malaria Research, Indian Council of Medical Research, New Delhi, India. Written informed consent was given by the parents and guardians of children before they were enrolled in the studies.

**Longitudinal cohort studies**. The Indian LCS study was conducted in Dumargarhi in the state of Jharkhand[22]. Briefly, out of 945 individuals (aged 1–82 years) that were enrolled in the study, 386 were sampled at cross-sectional survey-1 and were followed up actively and passively for malaria case detection for 13-months (April 2015 to April 2016). In the current study, we focused on 121 participants (aged 3–60 years) who were definitively exposed to *P. falciparum* malaria (Supplementary Table 1).

The African study was conducted in Asutsuare, Damgbe, West District, in Ghana[19]. In total, 798 children under 12 years were enrolled in May 2008. Venous blood was obtained at enrollment and children were followed up actively and passively for malaria detection for 42-weeks. Here, we have used available samples from 140 children (aged 1–12 years) (Supplementary Table 1). At both sites, febrile malaria was defined as any *P. falciparum* parasitemia confirmed by microscopy of stained thick and thin blood smears plus reported fever or axillary temperature ≥37.5 °C at the time of the visit. Individuals who suffered at least one case of febrile malaria during follow-up were considered susceptible (India, 50 [41.3%]; Ghana, 81 [57.9%]), while those who did not experience any episodes of febrile malaria despite having parasites at baseline were considered protected (India, 71 [58.7%]; Ghana, 59 [42.1%]).

**Leukocyte purification**. Peripheral blood leukocytes (PBLs) were isolated from whole blood samples from Danish donors by centrifugation at 800 g for 25 min. The layer containing leukocytes was harvested and the contaminating red blood cells were lysed by mixing with 10 parts of lysis buffer (155 nM NH$_4$Cl, 10 mM KHCO$_3$, and 0.1 mM EDTA; pH 7.4) and incubating for 10 minutes. After washing once with PBS, leukocytes were resuspended in cell medium (RPMI-1640 supplemented with 10% fetal bovine serum, 25 mM HEPES, 4 mM L-glutamine, and 25 μg/ml gentamicin). Cells were counted using a hemocytometer and diluted with cell medium to a concentration of $6 \times 10^5$ cells/ml.

For neutrophil purification, samples from healthy donors were carefully layered on top of an isotonic Percoll solution with a density of 1.077 g/ml and centrifuged at 800 g for 25 min. The leukocytes were resolved into two distinct bands. The upper layer containing monocytes and the lower one containing neutrophils. The contaminating red blood cells were lysed as described above. Neutrophils were further purified using the EasySep Human Neutrophil Isolation Kit (StemCell Technologies) following the manufacturer's instructions. Purified cells were resuspended in cell medium, counted, and diluted with cell medium to a concentration of $6 \times 10^5$ cells/ml.

**Parasite culture and merozoite isolation**. *P. falciparum* strain NF54 was cultured at 4% hematocrit in parasite growth medium (RPMI-1640 supplemented with 25 mM HEPES, 4 mM L-glutamine, 5 g/l AlbuMAX, 0.02 g/l hypoxanthine, and 25 μg/ml gentamicin). The culture was synchronized by treating with 5% sorbitol for 10 min. Mature trophozoites/early schizonts were purified with a magnetic separation unit and cultured further. The merozoites were obtained by filtering mature schizonts using a 1.2 μm filter. After removing hemozoin, merozoites were stained with 10 μg/ml of ethidium bromide for 30 min and washed twice with fluorescence-activated cell sorting (FACS) buffer (PBS with 0.5% BSA + 2 mM EDTA). Merozoites were opsonized with plasma for 30 min before addition to the phagocytosis assays. A 1:8,000 dilution of the plasma was used throughout unless stated otherwise. This dilution corresponds to a merozoite-IFA titer of 40,000, and it also corresponds to the higher antibody concentration at which neutrophils were found to be the dominant phagocyte. We also used further serial dilutions of plasma with a lower merozoite-IFA titer of 10,000 at which monocytes were found to be the major phagocytic cell.

**Phagocytosis assay**. Leukocytes were transferred to 96-well U-bottom plates containing $6 \times 10^4$ cells in 100 μl of cell medium per well. Opsonized, ethidium bromide-stained merozoites ($4–6 \times 10^5$ per well) were washed twice with FACS buffer, resuspended in cell medium, and 100 μl were added to each leukocyte containing well. After an incubation of 30 min (unless stated otherwise) at 37 °C and 5 % CO$_2$, plates were centrifuged in a prechilled centrifuge and washed twice with ice-cold FACS buffer to stop phagocytosis. Cells were resuspended in 200 μl of cold FACS buffer and incubated for 1 hour at 4 °C with 1:1600 FITC antihuman CD14 (clone TuK4; Thermo Fisher Scientific MA1-82074), 1:400 BV786 antihuman CD16 (clone 3G8; BD Biosciences 563690), 1:800 APC antihuman CD45 (clone HI30; BD Biosciences 555485), and 1:800 BV421 antihuman CD66b (clone G10F5; BD Biosciences 562940) antibodies. After washing thrice with FACS buffer, sample fluorescence was quantified with a CytoFLEX S (Beckman Coulter Life Sciences). Phagocytosis was determined by measuring the ethidium bromide fluorescence using the 610/20 nm detector. Data analysis were performed with Kaluza Analysis Software version 2.1 (Beckman Coulter Life Sciences). The gating strategy used is shown in Supplementary Figure 2.

**Blocking of Fcγ receptors**. Previous to the incubation with opsonized merozoites, cells were incubated for 30 min at 37 °C in cell medium with 10 μg/ml of antihuman CD16 (clone 3G8; BD Biosciences 555404), antihuman CD32 (clone FLI8.26; BD Biosciences 555447), antihuman CD64 (clone 10.1; BD Biosciences 555525), or an isotype control to block Fcγ receptors 3, 2, and 1, respectively. To block all surface-exposed FcγRs a combination of anti-CD16, anti-CD32, and anti-CD64 antibodies were used. After the incubation period, merozoites were added and the experiment continued as described above. Phagocytosis values for treated cells are presented relative to the value of untreated cells incubated with the same immune plasma.

**Merozoite flow cytometry-based immunofluorescence assay**. The amount of IgG deposited on the merozoite surface was determined by flow cytometry-based immunofluorescence assay (merozoite-IFA) as described in detail[20]. Median fluorescence intensity values are reported.

**Statistics and Reproducibility**. The Wilcoxon signed-rank test was used to evaluate differences between two groups of paired observations. The Friedman test with Dunn's multiple comparisons test was applied to estimate differences between three or more paired groups. Generalized estimating equations were utilized to account for paired observations in generalized linear models. Association between time-to-first febrile malaria episode and the categorized levels of opsonic phagocytosis by neutrophils or monocytes was analyzed by age-adjusted Cox-regression models. The correlations between opsonic phagocytosis and ADCI activity were

assessed by the Spearman correlation coefficient (*r*). All statistical analyses were performed as two-sided tests. *P* values of less than 0.05 were considered significant. Statistical analysis was performed using Prism 9 (GraphPad Software, Inc.) and *R* version 3.6.2 with the package geepack[39].

**Reporting Summary**. Further information on research design is available in the Nature Research Reporting Summary linked to this article.

## Data availability
The data that support the findings of this study are available from the corresponding author upon reasonable request and pending agreement from relevant ethics committees for clinical data.

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

## Acknowledgements
We thank the study participants, their parents, and guardians for volunteering and making this work possible. This study was supported by the Danish Ministry of Foreign Affairs (DFC file no. 14-P01-GHA) and the Danish Council for Strategic Research (grant number 13127).

## Author contributions
A.G.S. and M.T. designed experiments. A.G.S., I.H.K., and S.H. performed experiments. S.S., M.K.D., M.H.D., and B.A. provided samples and clinical data. S.S. and M.T. prepared the first draft. All authors contributed to discussing the data and proofreading the manuscript.

## Competing interests
The authors declare no competing interests.
