## [Transparent Peer Review File · Communications Biology]

Reviewers' comments:

Reviewer #1 (Remarks to the Author):

In this study, the authors developed an in vitro opsonic phagocytosis assay using peripheral blood leukocytes and clearly demonstrate that CD14++CD16- monocytes are the dominant phagocytic cells at very low antibody levels. They found that at higher antibody levels neutrophils are the main phagocytes in the opsonic phagocytosis of merozoites. In the case of monocytes mediated phagocytosis (FcγR) IIA plays a key role while neutrophils mediated phagocytosis FcγRIIIB acting synergistically with FcγRIIA. They also observed that opsonic phagocytosis activity by neutrophils is strongly associated with protection against febrile malaria in longitudinal cohort studies performed in Ghana and India and conclude that neutrophils are the main phagocytes of *P. falciparum* blood-stage merozoites.

This work is technically sound and the claims are supported by the experimental data. I recommend the consideration of this manuscript for publication.

Reviewer #2 (Remarks to the Author):

This is an interesting and well-designed work describing in detail the role of neutrophils and monocytes in the clearance of *P. falciparum* merozoites in vitro and in vivo. The in vitro experiments are carefully performed and analyzed. The data is well explained and supports the conclusions. The second part performing experiments with plasma samples of malaria patients, is an essential component, suggesting the in vitro results represent the real disease. This last part requires more clarifications, since the authors rely too much on previous descriptions without even giving a brief summary of some of the important data here. However, the results are solid and support the conclusions of the study.

Minor points:

In Fig. 3A the difference between anti-CD16 and anti-CD16+antiCD32 appears to be small in absolute numbers, although it is 61% proportionally it doesn't seem to be an important component of the phagocytic response. Error bars should be plotted for every condition relative to isotype control.

Line 156: This sentence is confusing since it doesn't explain that the phagocytic responses were measured in neutrophils from blood donors, not from Ghanaian children "...we used data and samples from a well-established longitudinal cohort survey (LCS) performed in Ghanaian children [19]. A wide range of phagocytic responses were observed for both neutrophils..."

Line 176: "Next, we repeated the phagocytosis assays using highly purified neutrophils (>99%) and merozoites opsonized with the same plasma". It is not clear what plasma are they referring to, the high phagocytosis-inducing plasma? All plasma in the cohort? Both cohorts? These data may be more clearly understood if presented in a table comparing both cohorts.

Line 194: the abbreviation ADCI is not spelled out in the manuscript.

Fig. 5 and the text show that there is no correlation between phagocytosis and ADCI. The title of this section states "IgG mediated merozoite-phagocytosis and ADCI, both independently correlate with protection against clinical malaria." However, this is not even mentioned in the text or in Fig. 5.

Reviewers' comments and our responses

Reviewer #1 (Remarks to the Author):

In this study, the authors developed an in vitro opsonic phagocytosis assay using peripheral blood leukocytes and clearly demonstrate that CD14⁺⁺CD16⁻ monocytes are the dominant phagocytic cells at very low antibody levels. They found that at higher antibody levels neutrophils are the main phagocytes in the opsonic phagocytosis of merozoites. In the case of monocytes mediated phagocytosis (FcγR) IIA plays a key role while neutrophils mediated phagocytosis FcγRIIB acting synergistically with FcγRIIA. They also observed that opsonic phagocytosis activity by neutrophils is strongly associated with protection against febrile malaria in longitudinal cohort studies performed in Ghana and India and conclude that neutrophils are the main phagocytes of *P. falciparum* blood-stage merozoites.

This work is technically sound and the claims are supported by the experimental data. I recommend the consideration of this manuscript for publication.

We thank the Reviewer for acknowledging our work and recommending the consideration of this manuscript for publication.

Reviewer #2 (Remarks to the Author):

This is an interesting and well-designed work describing in detail the role of neutrophils and monocytes in the clearance of *P. falciparum* merozoites in vitro and in vivo. The in vitro experiments are carefully performed and analyzed. The data is well explained and supports the conclusions. The second part performing experiments with plasma samples of malaria patients, is an essential component, suggesting the in vitro results represent the real disease. This last part requires more clarifications, since the authors rely too much on previous descriptions without even giving a brief summary of some of the important data here. However, the results are solid and support the conclusions of the study.

We thank this Reviewer for appreciating our work and highlighting some important points which have been addressed now. Please see our point by point response below. Necessary modifications have been introduced in the revised manuscript.

Comment	Response
1) In Fig. 3A the difference between anti-CD16 and anti-CD16+antiCD32 appears to be small in absolute numbers, although it is 61% proportionally it doesn't seem to be an important component of the phagocytic response. Error bars should be plotted for every condition relative to isotype control.	Figure 3 has been modified to show the comparison of every condition relative to the respective isotype control. Figure 3 legend has been modified accordingly.

	a Relative phagocytosis (%) IP, isotype, aCD16, aCD32, aCD64, aCD16+aCD32 b Relative phagocytosis (%) IP, isotype, aCD16, aCD32, aCD64, aCD16+aCD32
2) Line 156: This sentence is confusing since it doesn't explain that the phagocytic responses were measured in neutrophils from blood donors, not from Ghanaian children "...we used data and samples from a well-established longitudinal cohort survey (LCS) performed in Ghanaian children [19]. A wide range of phagocytic responses were observed for both neutrophils..."	Figure 3 PBLs and purified neutrophils were obtained from Danish blood donors. This has been clarified throughout the manuscript. Lines 156-157 "The PBLs used here in phagocytosis assays were obtained from a Danish blood donor." Lines 178-180 "Next, we repeated the phagocytosis assays using highly purified neutrophils (>99%; obtained from a Danish blood donor) and merozoites opsonized with the same Ghanaian plasma samples (n=140) [19] (Supplementary Table 1)." Lines 187-189 "In general, neutrophils in PBL preparations (obtained from a Danish blood donor) showed higher phagocytosis activity than monocytes (Figure 4c)." Lines 341-342 "Peripheral blood leukocytes (PBLs) were isolated from whole blood samples from Danish donors by centrifugation at 800 g for 25 min" Line 465-468 "Paired aligned dot plots coupled to box and whiskers plots showing the number of EtBr-positive neutrophils and monocytes present in PBLs from Danish blood donors which were used as effector cells to test the phagocytosis mediating ability of antibodies present in Ghanaian (a) and Indian (c) cohort plasma samples."
3) Line 176: "Next, we repeated the phagocytosis assays using highly purified neutrophils (>99%) and merozoites opsonized with the same plasma". It is not clear what plasma are they referring to, the high phagocytosis-inducing plasma? All plasma in the cohort? Both cohorts? These data may be	This experiment involving purified neutrophils used all (n=140) available plasma samples from the Ghanaian cohort. To clarify the number and origin of plasma samples used in immune epidemiology studies, participants

more clearly understood if presented in a table comparing both cohorts.	from the two cohorts have been describe in a new table (Supplementary Table 1). This has led to the following modifications to the manuscript: Lines 153-156 “To investigate the association between antibody-mediated merozoite-phagocytosis by human PBLs and protection against febrile malaria, we used clinical data and plasma samples (n=140) from a well-established longitudinal cohort survey (LCS) performed in Ghanaian children [19] (Supplementary Table 1).” Lines 178-189 “Next, we repeated the phagocytosis assays using highly purified neutrophils (>99%; obtained from a Danish blood donor) and merozoites opsonized with the same Ghanaian plasma samples (n=140) [19] (Supplementary Table 1). Children in the high purified neutrophil-phagocytosis group had a significantly (uHR = 0.43; 95% CI, 0.28–0.68; P < 0.001 and age-adjusted (aHR) = 0.47; 95% CI, 0.30–0.75; P = 0.001) higher probability of remaining free of malaria during the study period than those in the low purified neutrophil-phagocytosis group. Thus, supporting the importance of neutrophils in parasite-killing mechanisms. To further extend these findings and to strengthen the finding that neutrophils are important for NAI against malaria, we comparatively assessed plasma samples (n=121) in PBL-phagocytosis assay and used clinical data from a LCS conducted in India [21, 22] (Supplementary Table 1). In general, neutrophils in PBL preparations (obtained from a Danish donor) showed higher phagocytosis activity than monocytes (Figure 4c).”
4) Line 194: the abbreviation ADCI is not spelled out in the manuscript	This has been corrected. Lines 200-201 “The antibody-dependent cellular inhibition (ADCI) assay measures the parasite killing capacity of antibodies in collaboration with human monocytes.”
5) Fig. 5 and the text show that there is no correlation between phagocytosis and ADCI. The title of this section states “IgG mediated merozoite-phagocytosis and ADCI, both independently correlate with protection against	We agree. The title for this section has been changed to better reflect the text and Figure 5: Lines 198-199 “Relationship between antibody-mediated merozoite-phagocytosis by PBLs and

clinical malaria.” However, this is not even mentioned in the text or in Fig. 5.

antibody-dependent cellular inhibition activity in Ghanaian children.”

REVIEWERS' COMMENTS:

Reviewer #1 (Remarks to the Author):

All the concern raised by the second reviewer's is answered in the revised manuscript.

Reviewer #2 (Remarks to the Author):

The authors have addressed all the concerns.

Reviewers' comments and our responses

Reviewer #1 (Remarks to the Author):

All the concern raised by the second reviewer's is answered in the revised manuscript.

Reviewer #2 (Remarks to the Author):

The authors have addressed all the concerns.

We thank the Reviewers for acknowledging our work and recommending the consideration of this manuscript for publication.